# Characterizing sea ice melt pond fraction and geometry in relation to surface morphology

Lena G. Buth<sup>1</sup>, Thomas Krumpen<sup>1</sup>, Niklas Neckel<sup>1</sup>, Melinda A. Webster<sup>2</sup>, Gerit Birnbaum<sup>1</sup>, Niels Fuchs<sup>3</sup>, Philipp Heuser<sup>4</sup>, Ole Johannsen<sup>5,6</sup>, and Christian Haas<sup>1,7</sup>

Correspondence: Lena G. Buth (lena.buth@awi.de)

Abstract. Melt ponds play a crucial role in modulating the energy balance of Arctic sea ice by reducing surface albedo. While Arctic sea ice is becoming younger and smoother, this raises questions about how these changes affect melt pond characteristics, as the effect of ice deformation features, such as pressure ridges, on pond development on large spatial scales remains insufficiently understood. Here, we analyze 70 km² of high resolution airborne optical imagery and coincided coincident laser altimeter data from three research flights north of Greenland to investigate the relationship between ridge fraction and melt pond properties. Our results reveal that high-melt pond fractions are not exclusive to smooth ice but can also occur on heavily deformed multi-year ice can reach values comparable to those on smooth ice, with similar distributions observed for both. Furthermore, we find that ridge fraction influences both the size distribution and geometry of melt ponds on various typical ice types. Whether ridges constrain pond geometry in a way that increases or decreases pond shape complexity depends on pond size: small ponds are on average more complex in the presence of ridges, whereas large ponds are restricted in their complexity. This shift in behavior occurs around the characteristic size scale of 10² m² that coincides with the transition in pond fractal dimension. These results demonstrate the role of ice morphology in shaping melt pond characteristics and provide valuable insights for improving melt pond parameterizations in sea ice models.

#### 1 Introduction

Melt ponds are pools of meltwater that form on sea ice during the summer as a result of snow and ice melt. They are a critical component of the Arctic marine environment during the melt season as they substantially influence the energy and mass balance of sea ice. Melt ponds have a markedly lower albedo than the surrounding sea ice and snow surface (Perovich et al., 2002a). This contrast contributes to the ice-albedo feedback, a positive feedback mechanism in which the reduced surface albedo of melt ponds increases the absorption of solar radiation. This enhanced energy input accelerates surface melting, further reducing the albedo and reinforcing the feedback (Perovich and Tucker, 1997; Fetterer and Untersteiner, 1998). The increased energy

<sup>&</sup>lt;sup>1</sup> Alfred-Wegener-Institut, Helmholtz-Zentrum für Polar- und Meeresforschung, Bremerhaven, Germany

<sup>&</sup>lt;sup>2</sup>Polar Science Center, University of Washington, Seattle, USA

<sup>&</sup>lt;sup>3</sup>Institute of Oceanography, University of Hamburg, Hamburg, Germany

<sup>&</sup>lt;sup>4</sup>Helmholtz Imaging, Deutsches Elektronen-Synchrotron DESY, Hamburg, Germany

<sup>&</sup>lt;sup>5</sup>Division of Medical Image Computing, German Cancer Research Center (DKFZ), Heidelberg, Germany

<sup>&</sup>lt;sup>6</sup>Helmholtz Imaging, German Cancer Research Center (DKFZ), Heidelberg, Germany

<sup>&</sup>lt;sup>7</sup>Institute of Environmental Physics, University of Bremen, Bremen, Germany

absorption caused by melt ponds not only accelerates surface melting, but also contributes to sea ice thinning (Popović and Abbot, 2017). In addition, melt During winter, their influence continues as refrozen ponds inhibit sea ice growth in winter by forming an insulating ice lid. The pond water beneath this lid retains latent heat, which slows ice formation at the base of the underlying sea ice until the pond fully refreezes (Flocco et al., 2015). Beyond their impact on the physical properties of sea ice, melt ponds also play a role in the Arctic ecosystem by increasing light transmittance through the ice, which supports primary productivity (Frey et al., 2011; Nicolaus et al., 2012; Arrigo et al., 2012).

The timing and onset of pond formation depend on environmental conditions that trigger and sustain melting. Synoptic weather events transporting warm, moist air into the Arctic play a critical role in initiating pond formation by enhancing sensible and latent heat fluxes, which drive snow and ice melt (Skyllingstad and Polashenski, 2018). The formation of ponds is then regulated by the availability of meltwater and the permeability of the ice (Eicken et al., 2002).

Premelt The location of initial pond formation is determined by pre-melt surface height, especially by the presence of snow dunes. These represent elevated features on the sea ice surface, and as snow and ice begin to melt, determines where meltponds form, with ponds preferentially developing in meltwater preferentially accumulates in the lower-elevation areas (Barber and Yackel, 1999; I . However, rather than larger-scale topography, local depressions primarily control pond formation by dictating where meltwater accumulates (Landy et al., 2014). On multi-year ice, the topography inherited from previous melt seasons also influences pond formation (Eicken et al., 2004). Sea between them. These local depressions control the initial distribution of meltwater and thus determine the spatial pattern of early pond formation (Barber and Yackel, 1999; Polashenski et al., 2012; Petrich et al., 2012) . Similarly, sea ice pressure ridges constrain the lateral spread of meltwater, and on. Rather than being driven by larger-scale topography, pond formation is primarily governed by such small-scale surface roughness features (Landy et al., 2014). The detailed relationship between ice surface morphology and melt ponds differs across ice regimes, and most studies focus on one ice type only. On first-year ice, their presence limits the presence of pressure ridges has been shown to limit pond coverage (Nasonova et al., 2017). Consistently, Landy et al. (2014) observed a higher peak pond coverage on smooth landfast first-year ice compared to the previous year when the surface was rougher. On multi-year ice, however, Nasonova et al. (2017) found no strong correlation between surface roughness and the melt pond fraction. Still, topography influences pond formation on multi-year ice as well. Inherited topography from previous melt seasons influences pond formation, with some melt ponds reappearing in the same locations as in the previous year (Eicken et al., 2001). These low-lying areas often coincide with thinner ice, and their locations can be predicted from winter surface temperature anomalies and pre-melt elevation data (Thielke et al., 2023; Fuchs et al., 2025).

40

Melt ponds play a critical role in sea ice evolution, yet their representation in sea ice models remains challenging. If ponds are not included, ice thickness and volume can be overestimated, with simulations showing up to 40% higher sea ice volume at the September minimum compared to models that incorporate a melt pond scheme (Flocco et al., 2012). However, current parameterizations, such as the level-ice melt pond scheme in Version 2 of the Community Earth System Model (CESM2) and the melt pond distribution conservation equation in the Marginal Ice Zone Modeling and Assimilation System (MIZMAS), tend to overestimate the pond fraction pond fraction compared to observations (Webster et al., 2022). Since individual melt ponds are subgrid-scale features, they cannot be explicitly resolved in large-scale models. At the same time, the sensitivity of

model results to the melt pond parameterization is very high (Driscoll et al., 2023), highlighting the need for better process understanding to improve their representation.

As the frequency of pressure ridges has decreased on Arctic sea ice(Krumpen et al., 2025), the relationship between melt ponds and ridges and its implication for the albedo feedback (Landy et al., 2015) and under-ice solar partitioning (Katlein et al., 2016; Horvat et al. become subjects of increasing interest(Krumpen et al., 2025; Landy et al., 2015; Katlein et al., 2016; Horvat et al., 2020).

The objective of this study is to examine how the presence of ridges influences melt pond characteristics, utilising utilizing high-resolution airborne data collected during Arctic field campaigns over sea ice, covering a variety of different ice types and deformation grades during the melt season. By analyzing simultaneous camera imagery and laser altimeter measurements, we quantify melt pond fraction, size distribution, and shape complexity across different ridge conditions. These observations provide a detailed assessment of the relationship between ridges and melt pond properties, based on a large-scale dataset, thereby establishing a foundation for refining representations of melt ponds in sea ice and climate models.

### 2 Data and methods

# 2.1 Measurement flights

The data analyzed in this study were collected during the IceBird summer campaigns of 2016 (Krumpen et al., 2024a) and 2018 (Krumpen et al., 2024b), which were conducted using the Polar 6 research aircraft. IceBird is a long-term monitoring program designed to survey Arctic sea ice, with a primary focus on ice thickness measurements, using a tethered electromagnetic (EM) sensor, the EM-Bird. The survey flights and data are described in Belter et al. (2021). In this study, we focus on analyzing simultaneous altimeter and camera data, covering over 70 km² in total. Three flights north of Greenland, see Figure 1, were selected based on instrument performance, favorable weather conditions for optical imaging (e.g., absence of low clouds, fog, or atmospheric obstructions between the aircraft and the ice surface), and comparability in the seasonal context, with the aim of reducing the impact on pond evolution drivers other than ridges. From here on, we refer to the survey flights on 1 August 2018, 29 July 2016, and 1 August 2016 as the Eastern. Central, and Western flights, respectively, based on their relative positions. Table 1 provides a summary of the flight details. As the flights were conducted at low altitudes, the images do not overlap, but capture snapshots of the surface every few hundred meters. Altitude variations during the measurements were minimal, mostly within 2.5 m, with maximum variations of about 5 m. These differences in altitude, combined with the together with the fixed camera specifications, result in a varying affect the ground sample distance (GSD), defined as the distance between adjacent pixel centers on the ground, which in turn determines the surface area covered by each image. As a result, the surface area per image varies slightly throughout and, more noticeably, between the flights.

In the following subsections we describe the image processing including brightness correction, removal of the EM-Bird tow cable from the images, georeferencing and orthorectification of the images as well as the surface classification. Particularly the subsections In particular, Section 2.2 on brightness correction and Section 2.3 on the removal of the tow cable introduce novel approaches and are therefore described in detail. Readers seeking to apply these methods are encouraged to explore these subsections further, while others may proceed directly to subsection Section 2.6.

Figure 1. Map showing the flight tracks (circles) of all three survey flights, here named the Western (W), Central (C), and Eastern (E) flightflights, as well as along with the satellite-derived ice drift trajectories over for the preceding—70 days prior to each survey (Krumpen, 2018). The color scale represents—Trajectories are shown as semi-transparent dots, with one dot per day. Flight tracks and the corresponding drift paths are color-coded by the average number of days of continuous melt for days along each summer trajectory until the survey time, ealculated using based on the NASA Arctic Sea Ice Melt product of the respective year (2018 for W, 2016 for C and E).

# 2.2 Image brightness correction

A digital single-lens reflex (DSLR) camera with a focal length of 14 mm is mounted at the bottom of the aircraft. It captures RGB (red, green, blue) optical images of the sea ice surface. One typical raw image per flight is shown in Figure 2a, d and g. Brightness variations throughout a single flight arise from changes in atmospheric conditions, such as cloud cover or solar angle. Consequently, a brightness correction is necessary to ensure the comparability and consistency of images across varying light conditions during the flight. This correction ensures that a unified classification algorithm can be applied to the image set and optimizes the brightness contrast within each image for subsequent analysis. Typically, brightness differences occur gradually over entire flight lines and locally due to cloud shadows. Contrary to Webster et al. (2015), we did not experience such local shadowing and could thus omit manual intervention and create an automatic, more efficient correction algorithm that focuses on gradual difference.

**Figure 2.** Image processing steps: (a, d, g) Exemplary Examples of raw RGB image images from each flight, the cable of the tethered instrument is marked by a pink an orange outline. (b, e, h) Georeferenced and orthorectified brightness-corrected images, the instrument cable is removed. Pink circles Short horizontal lines in pink represent altimeter measurements, filled orange circles the thick green line where a ridge was detected. (c, f, i) Classified image with snow and ice in sage green, open water in dark blue and ponds in rosybrown.

Table 1. Overview of the acquisition parameters and sea ice conditions at the time of the three flights.




|                                           | Western Flight       | Central Flight              | Eastern Flight       |
|-------------------------------------------|----------------------|-----------------------------|----------------------|
| Date                                      | 1 August 2018        | 29 July 2016                | 1 August 2016        |
| Number of images                          | 3224                 | 665                         | 1578                 |
| Typical Mean image size                   | 143.7 m x 95.6 m     | 131.2 m x 87.3 m            | 131.2 m x 87.3 m     |
| Typical Mean ground sample distance (GSD) | 0.037 m              | 0.034 m                     | 0.034 m              |
| Total surveyed area                       | 44.5 km <sup>2</sup> | $7.6\mathrm{km}^2$          | 18.1 km <sup>2</sup> |
| Average flight altitude                   | 71.8 m / 236 ft      | 65.8 m / 216 ft             | 65.8 m / 216 ft      |
| Mean sea ice concentration                | 0.83 0.93            | <del>0.67</del> <u>0.83</u> | 0.720.91             |
| Estimated mean ice age (IceTrack)         | 3.1 yr               | 2.7 yr                      | 2.0 yr               |
| Mean days since melt onset                | 36                   | 51                          | 47                   |
| Mean melt pond fraction                   | 0.13                 | 0.28                        | 0.24                 |
| Mean ridge fraction                       | 0.14                 | 0.19                        | 0.13                 |

Previous work by Fuchs (2023a) and Neckel et al. (2023) implemented an empirical line method for brightness correction. We build our correction on their method by introducing updates that account for dynamic light conditions within a single flight: In order to account for changing light conditions, a brightness value is calculated for each image by determining the modal value of the sum of the R, G, and B Red, Green, and Blue channel brightness per image (modal RGB sum), as visualized in Figure 3. By separating the individual images, errors are less probable to accumulate as in the correction approach by Pedersen et al. (2009) that compares brightness on overlapping areas in images. In the dataset presented here, the most common surface type is relatively smooth snow or ice; thus, the modal RGB sum typically represents this surface type. However, the RGB sum of each surface type varies significantly along each flight, depending on the prevailing brightness conditions. For instance, values for snow and ice during the Western flight range from below 20,000 to over 50,000 (see Fig. 3). Given this variability, there is a need for a relative rather than a fixed reference. That is why the following approach does not rely on a single expected brightness value for snow or ice, but instead dynamically adapts to observed conditions and thereby ensures consistent correction across a broad spectrum of light conditions. Assuming the true color of snow and ice remains constant throughout the flight, the observed values of the modal RGB sum can indicate brightness conditions during image acquisition. This assumption based on a reference surface is similar to Renner et al. (2013), who corrected their images with the illumination of the EM-Bird in the images. Our assumption holds as long as most images are dominated by snow or ice. Images dominated by open water or melt ponds are filtered out from the brightness assessment step, as explained below, but it. These images are not discarded but are assigned a brightness value interpolated from neighboring images and are fully retained for all subsequent processing steps. It should be noted that the method is unsuitable this brightness correction method is not well suited for regions with many such images, such as the marginal ice zone (MIZ) with low sea ice concentration.

**Figure 3.** Brightness values of images captured sequentially during the Western flight. The x-axis represents the image number, while the y-axis shows the modal RGB sum for each image. Bright green markers indicate values identified as snow/ice, while blue markers denote outliers excluded from the analysis. The thick, color-coded <u>circles represent line represents</u> the smoothed snow/ice values, with colors corresponding to brightness quantile classes as indicated by the color bar.





A rolling maximum of the modal RGB sum is computed over 30 image windows to establish a baseline for expected values on a snow and ice surface. Outliers are identified as instances where the modal RGB sum is less than the rolling maximum divided by a factor of 2.5. This empirically determined approach ensures the capture of rapid but gradual changes in brightness while effectively filtering out open water surfaces. Residual data points, now consisting exclusively of images dominated by snow and ice, are smoothed with a rolling mean. Thereby, gaps introduced during outlier filtering are also filled, ensuring continuous data for brightness categorization. Each image is now assigned to one of 10 brightness categories, based on quantiles. Figure 3 illustrates the process of assigning brightness values and categories to each image. It shows the modal RGB sum for each image along the flight track, with the values used for brightness assessment in green and the outliers in blue, as well as the smoothed values (varying color). The horizontal lines indicate the boundaries of the 10 brightness categories. For each of the 10 brightness categories, a representative image is selected. Using a graphical user interface (GUI), the operator identifies dark (open water) and bright (snow/ice) areas within the selected images, as described in Neckel et al. (2023). The corresponding pixel values are extracted and then used to calibrate the brightness adjustment. Using these 10 images, a linear regression model is established to relate the brightness value (modal RGB sum) to the pixel values of snow/ice and open water, comparable to the histogramm histogram stretch approach proposed by Wright and Polashenski (2018). This model is then applied to all images, ensuring that dark areas such as open water correspond to a reflectivity of 0.1 and bright areas to 0.9 (Fuchs, 2023a). Fully linear sensor sensitivity is required for this conversion, which means that pixel values are directly linearly dependent on the brightness of the observed area. This has been confirmed in several studies for the applied camera system (Ehrlich et al., 2012; Carlsen et al., 2020; Fuchs, 2023a). The corrected RGB values for each image are thus calculated, enabling consistent classification across the dataset. Resulting brightness-corrected images can be seen in Figure 2b, e and h. We The described method ensures that the selected images used for calibration span the full range of brightness conditions encountered during a flight. As each image is assigned to one of 10 brightness categories, we determined that it was is sufficient to identify individual reference areas for each brightness class once reference areas in just one representative image per category, resulting in a total of 10 images assessed per flight. This revised approach significantly reduced the manual labor involved in analyzing optical imagery from long flights over Arctic sea ice, relying solely on the images themselves.

# 2.3 Removal of EM-Bird tow cable




In order to quantify melt pond characteristics, we first classify the surface types visible in the aerial images. Before performing this classification, however, we must address an issue caused by the EM-Bird and its cable, both of which are regularly visible in the camera's footprint, as shown in Figure 2. The EM-Bird itself, while frequently present, poses little problem due to its compact geometry. The cable, however, is more problematic due to its elongated shape and its potential to interfere with object segmentation after classification. Although the cable covers only a small portion of each image, its shape and variable appearance under different lighting conditions lead to inconsistent classification results: If classified as ice, the cable can artificially divide melt ponds into smaller, disconnected segments, biasing metrics such as pond size, perimeter, and shape complexity. If classified as a pond, the cable can act as a bridge between separate ponds, artificially connecting them into a single larger pond and introducing a positive bias in pond size and shape metrics. These effects make the cable a source of error in object segmentation. To ensure accurate segmentation and classification, the cable is removed from the images in a two-step process. Firstly First, the cable is assumed to be a straight line in the images, which is approximately true for the majority of the visible cable. To detect the line, a kernel is slid over the image, followed by the application of probabilistic Hough Line Finding, as implemented in the Python library scikit-image (Van der Walt et al., 2014). A continuous line is then created using RANSAC Line Regression. A 30-pixel-wide mask is generated around this line, corresponding to approximately 1 m on the georeferenced image or about 1.2% of the image pixels. This ensures that the majority of the cable is masked, even if it is not perfectly straight. As a second step, the masked area is inpainted to allow for subsequent analysis of the images. Inpainting is performed using the LaMa (Resolution-robust Large Mask Inpainting with Fourier Convolutions) image inpainting method (Suvorov et al., 2021), which reconstructs the missing region while preserving the overall appearance and texture of the surrounding surface. Examples of the resulting images after cable removal and inpainting are shown in Figure 2. While the inpainting tool performs remarkably well in recognizing the surrounding environment, generating realistic pixel values, and thereby substantially improving object segmentation in later processing steps, it is important to note that the inpainted pixels do not represent real data. Image and mask files are available as stated in the data availability section.

## 2.4 Georeferencing and orthorectification

To ensure accurate scaling and spatial alignment, we use direct georeferencing with the Python package cameratransform (Gerum et al., 2019) to project the images onto the surface. The algorithm processes each image along with all available geolocation data from the GNSS/INS unit, including longitude, latitude, altitude above the DTU 21 Mean Sea Surface (Andersen et al., 2023), and the rotational angles (roll, pitch, and yaw) of the aircraft. Using this information, the images are projected onto

the ice or ocean surface. The georeferenced images are saved in EPSG 3413, the WGS 84 / NSIDC Sea Ice Polar Stereographic North coordinate system.

The camera timestamps are precise only to 1 only precise to one second, which introduces a potential mismatch between the image data and the GNSS/INS information. To address this issue, a two-step approach is employed. Initially, the data are matched with 1-second precision. Then, using surface profiles from the laser altimeter, the time offset for the camera is refined manually for each flightby visually comparing datasets. After determining the precise time offset, the. This is done by visually identifying and matching distinctive features, such as pressure ridges and smooth ice areas, that appear in both the altimeter data and the images. By comparing the spatial alignment of multiple such features, a spatial offset is determined, which can be converted to a temporal offset. The georeferencing algorithm is rerun using this refined time offset to improve the spatial accuracy. The results are verified by re-examining multiple locations along each flight track.

# 2.5 Surface classification and geometric considerations




A variety of sea ice surface classification algorithms for airborne images is available (e.g. Buckley et al., 2020; Wright and Polashenski, 2010). For this study, we implement the pixel-wise classification algorithm developed For the semantic classification of the images into sea-ice surface types, we use the pixel-based classification algorithm by Fuchs (2023a), which uses a random forest classifier. It was particularly developed for the applied camera systems and allows for classification of aerial images collected under clear- and overcast skies. To ensure compatibility with our dataset after the modified brightness correction (see Section 2.2), we expanded the original training dataset with pixels sampled from our images and retrained the model. This algorithm While a variety of sea ice surface classification for airborne images is available (e.g. Wright and Polashenski, 2018; Buckley et al., 2020), we selected this algorithm due to its tailored design for the imaging system used in our campaigns. It provides detailed surface classification, distinguishing between subclasses such as bright and dark melt ponds, shadows on snow/ice or ponds, and biological indicators. However, for our purposes, we focus on three main classes: snow/ice, melt ponds, and open water. Typical classified images can be seen in Figure 2c, f and i.

Following the classification process, neighboring pixels of the same class are combined into objects. To reduce noise, objects with a size smaller than 100 pixels, here corresponding to an area of approximately 0.1 m<sup>2</sup>, are filtered out, as applied by Huang et al. (2016) and Fuchs et al. (2024). This process ensures that the classification results are robust and relevant to the scale of our analysis, with the smallest detected ponds being around 0.1

While the minimum observable pond size is defined by this threshold, the maximum size is limited by the image dimensions. Due to the relatively low flight altitude required for sea ice thickness retrieval, the classified images are small, and larger ponds are increasingly likely to extend beyond image boundaries. As stated by Perovich et al. (2002b), the probability that a circular pond with radius R is fully contained within an image of width W and length L is given by:

$$p(R) = \frac{(L - 2R)(W - 2R)}{LW + 2LR + 2WR + \pi R^2} \tag{1}$$

Using this equation, we estimate that the probability of fully capturing a circular pond with an area of 50 m<sup>2</sup> in size. Typical elassified images can be seen in Figure 2c, f and i—a threshold that will be relevant relevant in our analysis— is about 76%

for the Western flight and 74% for the Central and Eastern flights. As pond size increases, this probability decreases, and larger ponds are more likely to be cut off at image edges. As a result, the dataset predominantly captures smaller ponds in full, while larger ones may only appear partially. In such cases, we treat ponds intersecting image borders as if they were whole. Consequently, our observations are inherently biased toward smaller ponds, and this constraint shapes the scope of our analysis, focusing on small to medium-sized melt ponds.

To evaluate pond shapes, it is essential to define the pond edges. However, the exact shape of these edges is not inherently defined, illustrating a variation of the "coastline paradox" (i.e., the measured perimeter depends on the resolution of the measurement, Mandelbrot (1967)). In this case, pond edges are determined by the classification algorithm, which defines boundaries along pixel edges, often resulting in step-like angular geometries. To obtain more realistic pond circumferences and reduce angularity, the edges are smoothed using the simplify function from the Python library GeoPandas (Jordahl, 2014). The smoothing process is applied with a tolerance equal to the pixel length, ensuring that the new, simplified edges remain within this tolerance value from the original boundaries.

# 2.6 Laser-based pressure ridge identification






Surface profile data is obtained from a single-beam laser altimeter mounted on the EM-Bird. The altimeter has an alongtrack resolution of about 0.5 m, a beam diameter of 6 cm and a sensor accuracy of 5 cm (Krumpen et al., 2025). It provides surface profile data in only one dimension, along the flight track, but can be used to identify pressure ridges on the surveyed ice. A point is counted as ridge, if it belongs to a feature reaching sail height of at least 0.6 m relative to the height of surrounding level ice. Such features may be composed of ice and/or accumulated snow; however, we refer to them as ridges throughout this study, as pressure ridges are the dominant elevated features on the sea ice during these summer surveys. While the threshold of 0.6 m accounts for the uncertainty of the ridge detection method, it captures only the higher end of the sail height spectrum. However, since ridge sail height distributions on sea ice follow a negative exponential function (Wadhams, 1980; Rabenstein et al., 2010), any chosen height threshold is suitable to distinguish rough and less rough ice. Uncertainties of reconstructed surface profiles are discussed in Krumpen et al. (2025). A cross-comparison of different laser scanning systems revealed that the towed laser mounted inside the EM-Bird can underestimate ridge density by up to 0.45 km<sup>1</sup> and ridge height by approximately 5 cm, while overestimating spacing by 2.25 m (see also Table 5.1 in Suhrhoff (2021)).

## 2.7 Linking ridge data with melt pond observations

The melt pond data are linked with ridge data derived from laser altimeter measurements based on geolocation. Ridge fraction per image is calculated using the laser altimeter points within the spatial extent of each classified image, see Figure 2b, e and h. It is defined as the number of points identified as part of a ridge feature divided by the total number of altimeter point measurements and the sea ice concentration in that image. In this case, the ridge fraction is a more representative metric than, for example, the also commonly used ridge frequency, as it takes into account the ridge width. For some analyses, this derived ridge fraction is directly used and linked to the ponds on the corresponding image. In other cases, the classified images were grouped into three ridge fraction categories: low, medium, and high, based on quantiles of the observed ridge fraction

distribution across all images. The This means the lowest third of the distribution, comprising ridge fractions below 3.9%, is categorized as "smooth ice", while the highest third, comprising ridge fractions above 17.5%, is categorized as "rough ice". Specifically, these thresholds represent the 33rd and 67th percentiles of the observed ridge fraction range, respectively. The analysis focuses on these two categories to emphasize the contrast between areas with minimal and substantial ridge presence. We acknowledge that ice with ridge fractions below 3.9% is not necessarily smooth in an absolute sense, especially given the 0.6 m threshold used for ridge identification. Surface roughness at smaller scales is still possible within this category. However, for simplicity and improved readability, we refer to these categories as "smooth" and "rough ice" throughout this study.

If derived this way, the ridge fraction per image is highly dependent on the orientation of the ridge as compared to the flight direction. In addition, due to the heterogeneous nature of the sea ice surface, a single one of our images is typically too small to provide a representative sample of pond statistics on the respective surface. To account for these effects of ridge orientation and surface variability, and to ensure a more robust comparison across the ridge fraction categories, we resample the data by grouping several images at a time. A sensitivity analysis not presented here has shown that five is a suitable number of images to reduce noise while preserving realistic variability in pond statistics. Consequently, for each group, five images are randomly selected from within the same ridge fraction category ("smooth" or "rough"). Statistics per group are retrieved by concatenating the list of properties of individual ponds per image and averaging over the ridge fraction per image. This way, we derive pond fraction, mean circularity and ridge fraction for these groups of randomly sampled images. This resampling method enhances the impact of ridge fraction by smoothing out local effects and reducing the influence of ridge orientation relative to the flight direction. Without resampling, the calculated ridge fraction for individual images could be biased by the alignment of ridges with the aircraft's path: if a ridge is predominantly oriented in one direction, an image capturing an ice segment could either over - or underestimate the ridge fraction depending on its alignment with the ridge. By randomly sampling multiple images per category, we mitigate this orientation bias and obtain a more representative estimate of ridge fraction and its relationship to pond characteristics.

### 2.8 Sea ice age and drift trajectories from Lagrangian drift model





To estimate the age of surveyed sea ice, we employed the Lagrangian drift model IceTrack (Figures 1 and A1). This system traces the midpoints of 10 km surface-profile segments — extracted from single-beam laser altimeter data in 2016 and 2018 (see Section 2.6) — backward in time. IceTrack, which has been widely used in various studies (e.g. Peeken et al., 2018; Belter et al., 2021) and is described in detail by Krumpen et al. (2021), computes trajectories based on the OSI SAF OSI-405-c motion product (Lavergne, 2016). The tracking procedure halts when the drifting ice either contacts the coastline or fast ice edge, or when the satellite-derived ice concentration (from OSI SAF OSI-430-a) drops below 15 %. The resulting ice age is determined from the duration of these backward trajectories. A broader evaluation of uncertainties in the sea ice trajectories, conducted by comparing model outputs with drifting buoys (Krumpen et al., 2020), indicates a relatively small deviation between observed and modeled tracks (60±24 km over 320 days).

## 2.9 Sea ice melt product

We use NASA's Arctic Sea Ice Melt product (Markus et al., 2009), based on SSM/I daily averaged brightness temperatures, to gain insight into the length of the melt season in the relevant regions and the respective years. The first day of continuous melt, i.e. the day from which on melt conditions were continuously observed until autumn freezing occurs, is provided as a gridded annual dataset. As this first day of melt is a stationary value that does not take the drift history of the surveyed ice into consideration, we do this retrospectively: Using the ice drift trajectories described in Section 2.8, Figure 1 shows the number of days of continuous melt that have elapsed up to the dates of the survey flights. This value is determined by averaging the number of days between melt onset and the flight date for each point along the 70-day summer drift path leading to the flight track location. Although this does not strictly correspond to the actual number of melt days along the drift path, it provides a more realistic representation of the melt history than simply the melt onset date at the flight track location.

### 3 Results and Discussion





## 3.1 Ice conditions during and prior survey flights

The satellite backtracking, visualized in Figure 1 (full trajectories and respective ice age in Figure A1 in the Appendix), reveals diverse origins and past ice conditions of the surveyed ice, which we can link to the observed visual characteristics of the ice surface. Using the drift trajectory information, Figure 4 shows the sea ice concentration of the surveyed ice along its drift path during the 200 days prior to the flight. Sea ice in the observed area included a variety of dominant ice types of the Arctic Ocean and distinct ice conditions, as summarized in Table 1 and described in detail below:

The Western flight primarily surveyed third-year ice (TYI), with some areas of second-year ice (SYI) and patches of fourth-year and older ice. Satellite backtracking shows the various origins of the ice, with part of it having been transported through the Beaufort Gyre. The surface appeared generally rough, with a high prevalence of ridges, yet large areas of level ice were also present (example shown in Fig. 2a). The surveyed region exhibited high ice concentration, with large uninterrupted floes, particularly toward the north. According to the backtracking analysis, the ice concentration was continuously high along the drift trajectory and recovered quickly after the only two lower ice concentration events in spring and summer, see Figure 4. The airborne camera data confirm that the sea ice concentration at the time and location of the Western flight was the highest of all flights. In fact, with a mean of 0.830.93, it was considerably higher than that observed during the Central and Eastern flightsflight. Compared to the other flights, the ice surface was brighter, much fewer-less melt had occurred, as evidenced by extensive snow dunes during parts of the flight and overall less slush. Melt dryer snow, Based on visual qualitative assessment of the images, melt pond conditions were diverse, with both bright and darker ponds. Darker ponds were predominantly found on level ice. Horizontal drainage channels were observed mainly on level ice, and in some regions, extensive pond networks were interconnected by narrow channels.

The Central flight (example shown in Fig. 2d) covered regions with a mix of ice types, including landfast first-year ice, second-year ice, and multi-year ice. The satellite backtracking shows that much, as inferred from satellite backtracking of the ice drift

(see Section 2.8 and Fig. A1). Much of this ice originated from the East Siberian Sea. The surveyed area featured numerous smaller floes, leading to an increased presence of submerged ice, here referring to parts of the ice floe that lie below the water surface, e.g. at floe edges, but are still visible in the nadir images. Sea ice concentration was the lowest among all flights, with a mean ice concentration of 0.67 0.83 derived from airborne imagery. The satellite observations show that the sea ice concentration had been decreasing almost continuously for approximately 50 days prior to the flight (see Figure Fig. 4). This flight was conducted in 2016, which was identified as a year with anomalously low modal sea ice thickness, potentially attributed to an Atlantification event with increased ocean heat flux (Belter et al., 2021). The observed low sea ice concentration coincides with this anomaly. Compared to the Western flight, ponds appeared less diverse in color and shape. Darker ponds were mainly found on level ice, and some had melted through. Unlike the Western flight, where ponds were often connected by narrow channels, many ponds in this region had merged into larger, continuous networks.

The Eastern flight predominantly surveyed second-year ice. The ice in this region followed the Transpolar Drift, with its origin traced back to coastal areas, in this case mostly the New Siberian Islands (see Fig. A1). The ice's coastal origin is also evident from the presence of sediments on the ice, as observed in the images. The general ice conditions were similar to those of the Central flight, with a predominance of level ice over ridged ice, occasional sediment deposits, and widespread slush-covered areas areas of water-saturated snow (example shown in Fig. 2g, e.g. roughly one-third up from the bottom of the image). Despite being conducted in a different year and region, the ice concentration prior to the flight was comparable to that of the Western flight. However, airborne observations revealed a considerably Airborne observations revealed only a slightly lower mean sea ice concentration of 0.720.91. Melt pond conditions varied along the flight track. At the beginning and end of the flight, individual and connected ponds were present, with bright-colored ponds occurring on ridges and darker ones on level ice. In the middle part of the flight, extensive networks of dark ponds and slushy surfaces water-saturated snow became more prominent.

Figure 1 shows the considerable timing difference in continuous melt onset across the regions surveyed during the three flights. For the Western flight, the much later melt onset, with some regions experiencing melt just two weeks before the overflight, suggests a distinct delay in melt processes compared to the earlier melt onset before the Central and Eastern flights.

# 325 3.2 Melt ponds along the flight track

Melt ponds are of critical importance in regulating the surface albedo and energy budget of sea ice. It is therefore essential to gain an understanding of the ice characteristics that control melt pond fraction. This subsection examines the relationship between the distribution of melt pond fractions and the presence of ridges on the sea ice.

## 3.2.1 Melt pond fraction



As shown in Figure 5, the observed melt pond fraction was varied throughout the flights. The Central and Eastern flights exhibit higher melt pond fractions, with mean values of 0.28 and 0.24, respectively, in regions where early melt onset was observed. In comparison, the melt pond fraction observed during the Western flight is lower, with an average of 0.13. This flight was conducted in a region with an overall later melt onset. While where the melt onset occurred later and closer in

**Figure 4.** Mean sea ice concentration and its standard deviation of the surveyed ice regions along their drift paths during the 200 days prior to the flight, derived from the OSISAF-OSI SAF sea ice concentration product. Crosses mark the mean sea ice concentration observed at the time of each flight, using the classified airborne images. For seasonal context, the green shaded area marks the annual maximum sea ice extent in early March.

time to the survey date, reducing the time available for pond formation and evolution before the overflight. However, the melt onset date alone does not determine the pond fraction at any given time. While it provides context for the visual differences in ponding during the three flights, its relationship to the observed melt pond fraction is not straightforwardadditional factors influence pond development. Throughout the four stages of melt defined by Eicken et al. (2002), melt pond coverage evolves continuously with periods of expansion, drainage, and potential refreezing. As a result, the melt onset date alone does not determine the pond fraction at any given time. The observed melt pond fractions in our data reflect this inconsistent complex temporal variability of ponds.




Figure 6a illustrates the distribution of pond fractions for smooth (orangegold) and rough (red) ridge fraction categories across the three flights. The pond fraction depicted herein represents the mean of each random sample of five images, as detailed in Section 2.7. The figure highlights that the variation in melt pond fraction between the three flights is more pronounced than the differences between ridge categories within each flight. In the Central flight, a slight difference between rough and smooth ice is evident, with rough ice generally exhibiting lower pond fractions. However, in the Eastern flight, the differences between the two categories are minimal, and the Western flight shows no clear distinction between pond fraction on rough versus smooth

Figure 5. Map of melt pond fractions per image along the flight tracks.

ice. These findings indicate that ridges are not the sole factor influencing melt pond fraction, as considerable ponding occurs even in ridged ice areas. We thus expect other factors to play a more important role in controlling pond distribution.

# 3.2.2 Pond fraction and ice age



Differences in pond characteristics depending on ice age were reported to be very highly variable. For instance, Buckley et al. (2020) observed higher melt pond fractions on first-year ice (FYI) compared to multi-year ice (MYI), while Wright et al. (2020) observed no difference in the mean pond fraction but a higher variability on FYI. In our dataset, we observed higher pond fractions during the Central and Eastern flights, where the ice surface predominantly appears more level, a characteristic often associated with younger ice. In contrast, lower pond fractions were observed during the Western flight, which displays signs of older, more deformed ice. However, the lack of substantial amounts of FYI limits our ability to make conclusive comparisons. The majority of the ice surveyed is second-year ice (SYI) or older, with only a small amount of landfast FYI encountered during the Central flight. While this FYI shows the highest melt pond fractions, this is based on a limited number of images (56). Although some differences in pond fraction between younger (FYI + SYI) and older (TYI and beyond) ice are observed, these differences are small and inconsistent across the flights. Moreover, the satellite-derived ice age data are

**Figure 6.** Distribution of (a) mean pond fraction and (b) mean pond circularity values of 5 resampled images, shown for smooth (<u>yellowgold</u>) and rough (red) ice categories. The width of the violin is normalized to be proportional to the number of observations. (c) Example pond shapes, including their circularity value C.

available at a 10 km grid resolution, meaning that each grid cell often represents a mixture of floes with varying ice ages, making the classification of ice age more indicative of the dominant ice type rather than of specific floe characteristics.

# 3.2.3 Discussion of seasonality and difference between the flights




In order to interpret the results presented here, it is necessary to acknowledge that the survey flights represent snapshots in both space and time. The varying results underscore the fact that no simple rules can be established, as the processes influencing melt pond coverage are many and complex. In the following, we will address selected aspects of these processes, with the understanding that this discussion is not exhaustive.

A major factor influencing the observed distribution of melt ponds is the progression of melt throughout the season, including the role of drainage processes. Melt ponds form in local depressions rather than exclusively at the lowest surface elevations on a larger scale (Landy et al., 2014). As a result, ponds develop on both level and rough ice. Later in the season, drainage processes increasingly shape the distribution of ponds (Webster et al., 2015). Pond persistence is determined by the proximity to drainage channels, as meltwater preferentially drains in areas where such channels are accessible (Landy et al., 2014). In our study, the timing of melt onset and the visible state of the ice indicate that the ice surveyed during the Central and Eastern flights in 2016 was at a more advanced melt stage than during the Western flight in 2018, despite the similar time of the year. Features such as horizontal drainage channels and melt holes were frequently observed in 2016, suggesting that most ponds, regardless of whether they formed on rough or smooth ice, had already undergone drainage. At this stage, ponds that remained

or re-emerged were hydraulically connected to the ocean, meaning their water level could not substantially rise above sea level, as any excess would rapidly drain (Popović and Abbot, 2017). In this case, new or reformed ponds would preferentially persist in lower-elevation areas, as water on more elevated ridged ice would quickly drain away. In contrast, such drainage features were less common in 2018 and were predominantly observed on level ice. Limited access to drainage channels on rougher ice would suggest relatively high pond fraction in these areas. This is in agreement with our observation during the Western flight, where almost no difference in pond fraction could be observed between smooth and rough sea ice surfaces.

Other than the later stage of melt, we also observed a lower sea ice concentration during the Central and Eastern flights compared to the Western flight (see Section 3.1). This lower concentration was associated with a higher number of floe edges, allowing for more ridging at these floe edges, presumably resulting from deformation caused by floe movement. With melt pond fraction being generally lower towards the floe edges (Wright et al., 2020; Webster et al., 2022), this introduces a negative bias to the pond fraction of these images with low sea ice concentration.

In addition, the observed lower sea ice concentration and the occurrence of more brash ice during the Central and Eastern flights allow for more misclassification of submerged ice as melt pond. However, the fraction of submerged ice remains small compared to the true melt pond fraction, and a clear distinction is beyond the scope of this study.

## 390 3.3 Pond geometry and size distributions





While the melt pond fraction and its connection with ridges play a critical role in determining the overall surface albedo and energy balance of sea ice, understanding the specific arrangement of these ponds is crucial for other processes. One key factor is the under-ice light distribution, which affects biological activity, including the growth and distribution of algae and other organisms beneath the ice (Arrigo et al., 2012; Horvat et al., 2020)(Arrigo et al., 2012). Therefore, not only the pond fraction but also the spatial characteristics, including the size and shape of individual ponds, influence these ecological and physical processes (Horvat et al., 2020). In this subsection, we examine how the distribution of melt ponds is related to the presence of ridges on the ice surface, investigating both the pond size distributions and shape characteristics in relation to ridge fraction.

### 3.3.1 Power-law distribution of pond sizes

The distribution of melt pond sizes observed across all analyzed images follows a power law, see Figure 7, consistent with previous studies on pond size distributions on Arctic sea ice (Perovich et al., 2002b; Kim et al., 2013; Huang et al., 2016; Popović et al., 2018). This power law can be expressed as:

$$p(x) \sim x^{-\alpha} \tag{2}$$

where the exponent  $\alpha$  determines the steepness of the distribution and quantifies the relative abundance of small versus large ponds. Typically,  $\alpha$  is positive and greater than 1. As  $\alpha$  increases, the number of smaller ponds becomes disproportionately higher compared to larger ones. Conversely, a lower  $\alpha$  indicates a more uniform distribution of pond sizes.

Figures 7a and 7b show the observed melt pond size distributions for different ridge fractions. As we observe no qualitative difference and only negligible quantitative differences between the three flights, the observational data presented here comprises

**Figure 7.** (a) Pond size distribution p(x) on a logarithmic x and y scale for all observed ponds (red), as well as only the ponds on images with smooth (green) and rough (yellow) ice. Shaded areas of these colors show the pond size distribution using a linear bin width, the solid line shows the same data using a variable bin width better adapted for power law distributions. Dashed lines represent the power law fit for data within the black rectangle. The area shaded in light blue shows the range of pond sizes for which the power law fit was applied. (b) As in a, but zoomed into the black rectangle, where the pond distribution is well represented by the power law function.

pond sizes from all flights, totaling over 1.2 million ponds. The shaded areas indicate the distributions using linear binning, while the solid lines show the same data with a variable bin width approach, implemented with the Python library powerlaw (Alstott et al., 2014), to better represent the data on the logarithmic x-axis. The dashed lines correspond to the fitted power-law curves. The power-law fit is applied exclusively to the pond size range between 1 and 50 m², marked in light blue and with a black rectangle, where the data exhibits a linear trend in the log-log representation. This linearity indicates that a power-law distribution is an appropriate model for the size distribution within this range.

At larger pond sizes, we observe a deviation from the power law fit. The curve dips, indicating the limitations of our dataset, which is constrained by the image coverage. The field of view in the aerial images restricts the maximum detectable pond size (which theoretically equals the image size), indicating that the dataset can only accurately capture the distribution of pond sizes up to a certain threshold.

Within the range where the power law applies, the resulting exponents are approximately 1.45 (R = 0.99) for all ponds, with  $\alpha$  values of 1.40 (R = 0.98) for ponds on smooth ice and 1.51 (R = 0.98) for ponds on rough ice, as illustrated in Figure 7b. The different exponents indicate that the pond size distributions vary between regions of smooth and rough ice. Smoother ice is thus associated with a smaller exponent  $\alpha$ , indicating a more uniform distribution of pond sizes, where the difference between the occurrence of small and large ponds is less pronounced. In contrast, rougher ice results in a steeper size distribution with a larger  $\alpha$ , implying a greater difference in the number of small versus large ponds. Thus, the more ridged the ice, the fewer large ponds appear relative to small ones, showing how ridges constrain the size and spread of melt ponds. Despite these differences,

Figure 8. (a) Binned pond size and pond perimeter data of all observed ponds, color coded using the mean ridge fraction of the images that the ponds in the corresponding bin appear on. Pink circles denote invalid bins, where the number of ponds in this bin is lower than the threshold of five. The red rectangles mark the excerpt shown in more detail in d. (b) Scatterplot of pond size and pond perimeter for all observed ponds. For better conceptual understanding, two red lines indicate the approximate perimeter-area ratio for different pond sizes. A shift in this ratio can be seen around 10<sup>2</sup> m<sup>2</sup>. The dashed blue line shows the perimeter-to-area ratio of a circle. (c) Conceptualized and simplified visualization illustration of relationship the general trend of decreasing ridge fraction with increasing pond size and perimeter, as seen in panel a. (d) Excerpts from a, showing the color coded ridge fraction in two different pond size bins for increasing pond perimeter (increasing pond shape complexity). Examples for ponds within these two size bins are given by the smallest and the largest pond shape in Figure 6c.

all the observed exponents fall within the range reported in previous studies on Arctic melt pond size distributions, further supporting the usability of our pond size data for subsequent analysis.

## 3.3.2 Influence of ridge fraction on pond shapes

Pond circularity is a valuable metric for characterizing the shape and complexity of melt ponds. It is defined as

430 
$$C = \frac{P^2}{A}$$
 (3)

where P represents the perimeter and A denotes the area of the pond. This measure quantifies how closely a pond's shape approximates a perfect circle: A value of  $C = 4\pi$  corresponds to a circular form; higher circularity values indicate increased complexity (Perovich et al., 2002b). It is important to acknowledge that the absolute values of circularity can be influenced by the method used to smooth pond edges and the degree of smoothing in relation to the pixel size. Therefore, direct comparisons of these absolute values across different studies may not yield meaningful insights. However Additionally, ponds that are only partially captured within an image may appear less geometrically complex than they actually are, as portions of their shape are not visible. This phenomenon is most evident in larger, more irregular ponds, where the removal of branches or extensions can lead to a reduction in the perceived complexity and in particular on the calculated circularity value. For ponds of smaller, rounded proportions, the impact is negligible. The ponds are already not perfectly circular due to the pixel-based classification process; therefore, an image edge does not significantly alter their shape. The intersection of ponds with image edges thus introduces a bias towards rounder ponds, corresponding to lower circularity values. Nevertheless, within the context of this study, the same methodology has been consistently applied and the same limitations apply, allowing for valid comparisons of circularity among ponds within our dataset.

In Figure 6b, we present a violin plot depicting the mean pond circularity across the three flights, categorized by ridge fraction.

The circularity values represent the mean calculated from all ponds within samples composed of five randomly selected images as described in section 2.7. Notably, in the Central and Eastern flights, ponds on smoother ice tend to exhibit a lower circularity value C (indicating a less complex shape) compared to those on rougher ice, although the difference is subtle in the Central flight. The Western flight shows the opposite behaviour behavior, with ponds on rougher ice appearing more circular.

To better understand the influence of ridges on pond shapes and to clarify the observed differences among the various flights,

450 we present Figure 8a, which. Panel a shows the distribution of melt pond areas and perimeters for every individual pond on
a binned grid. Each bin is color-coded to represent the corresponding mean ridge fraction. The bins represent melt ponds of
varying sizes and shapes, and their corresponding ridge fraction is averaged based on the images from which these originate.

To ensure the robustness of the following analysis, bins containing fewer than five ponds (pink circle) were excluded. Several
key trends emerge from this figure, each represented by one subsequent panel of this figure:

1. An increase in the perimeter-area perimeter-to-area ratio is observed with an increase in pond size, see the conceptual Figure 8b. The overall distribution of data curves upward, indicating a shift from simple to complex pond shapes as the pond area increases. This is consistent with previous findings on melt pond geometry. Larger ponds tend to exhibit disproportionately larger perimeters, which signals a shift in fractal dimension with increasing pond size at a critical length scale of about 10<sup>2</sup> m<sup>2</sup> (Hohenegger et al., 2012; Bowen et al., 2018).

460 2. A clear second trend is evident in Figure 8a and conceptually simplified visualized in Figure 8c, which illustrates that as the background mode of the dependency between pond geometry and surface roughness: As both pond area and perimeter increase, the ponds tend to appear on ice with, on average, lower ridge fractions. This shows that large ponds with complex geometry tend to be limited to regions of smoother ice. This observation is consistent with the differences in the exponent α found in the pond size distribution from the previous subsection in Section 3.3.1. Conversely, smaller ponds are more frequently observed in regions of rougher ice, where ice ridges are likely to exert a constraint on their size.

3. Ponds with higher perimeter-to-area ratios (i.e., more complex shapes) tend to form on more ridged ice, across a broad range of pond sizes, from the smallest detected ponds up to about  $10^2$  m<sup>2</sup>. This indicates that ridges may impede the natural pond formation, particularly in smaller, less complex ponds. On smoother ice, these ponds would likely assume a more rounded shape, whereas ridged ice constrains their expansion, resulting in more irregular and elongated perimeters. This is visualized in Figure 8d, showing an excerpt of Figure 8a for the ponds with an area of about 1 m. It is noteworthy that this trend is reversed for larger ponds, which tend to form more complex and interconnected shapes on smooth ice, see the excerpt for ponds with areas of about  $10^3$  m<sup>2</sup> in Figure 8d. This may be attributed to the fact that larger ponds spread over a broader area, where ridges exert a less pronounced influence on shape, would be required to fully capture this effect. The transition between these behaviors appears to occur at the same scale as the shift in fractal dimension reported in previous studies (e.g., Hohenegger et al., 2012), which is also reflected in the bend of our data shown in Figure 8b, though larger images -(i.e. higher flight altitudes-) would 475 be required to fully capture this effect. These findings highlight the complex relationship between pond size and ridge fraction, with pond circularity being influenced by the scale of the pond relative to the ridges. For pond sizes below a certain threshold (around 10<sup>2</sup> to 10<sup>2.7</sup> m<sup>2</sup>, close to the critical length scale of fractal dimension), ridges physically disrupt pond expansion, resulting in less circular ponds as ridges impede the formation of smooth edges. In contrast, for pond sizes above this threshold, 480 ridges play a different role by limiting the development of large, connected pond networks. This constraint actually results in larger ponds being more circular on ridged ice compared to ice with fewer ridges. The shift in fractal dimension mentioned in the first point is more evident on smooth ice (lighter colors in Figure 8a), where ponds are free to spread and develop more complex shapes. On ridged ice, ponds are less likely to reach this higher fractal dimension due to the physical constraints imposed by the ridges.

With these new insights, the contrasting pond geometries between the Central and Eastern flights and the Western flight shown in Figure 6b can be explained by differences in flight altitude: The Central and Eastern flights, conducted at a lower altitude, primarily captured smaller ponds, whereas the Western flight, flown at a higher altitude, observed larger, more connected structures. This transition across scales further supports the observed relationship between ridge fraction and pond geometry.

#### 3.4 Limitations and recommendations for future surveys

495

Our analysis provides valuable insights into the relationship between spatial melt pond properties and pressure ridges, yet certain limitations should be considered when interpreting the results.

An important limitation is the range of observed pond sizes. Compared to other studies, our dataset lacks very large ponds, which is primarily a result of the relatively low flight altitude necessary for electromagnetic (EM) ice thickness measurements. Flights at higher altitudes with overlapping image coverage would help capture a more representative pond size spectrum.

Additionally, if surface topography derived from photogrammetry or even pond depth is of interest, grid flights, as demonstrated by Fuchs et al. (2024), are recommended.

While our classification of melt ponds is generally reliable, some misclassifications occur due to submerged ice, brash ice, or ice edges falsely being identified as ponds. A detailed analysis of the classification accuracy for the used algorithm can be found in Fuchs (2023a). Additionally, our ridge detection method does not allow us to distinguish between snow and ice in the

detected surface features. As a result, elevated structures identified from the laser altimeter are referred to as ridges throughout this study, although they may in some cases include contributions from snow accumulation. This limitation should be kept in mind, as snow depth and distribution can influence early meltwater pooling and thereby the location and development of melt ponds.

The use of ridge fraction as a primary metric to assess ridge influence on melt ponds is useful but not ideal for capturing all aspects of the impact of ridges on pond formation and evolution. Ridge distance likely plays an additional role in shaping pond characteristics, and future studies could benefit from incorporating this parameter. In our study, this parameter is not accessible, due to the one-dimensional nature of the laser altimeter data. This limits the ability to directly measure distances between individual ponds and ridges, as ridge fraction is derived from a single transect across each image. In addition, for a more detailed spatial analysis, such as calculating these pond-to-ridge distances, a more precise co-registration of datasets would be required. The geolocation accuracy of our data is high but not pixel-precise, with potential offsets of a few meters. However, the large number of samples in our dataset ensures that our statistical approach remains robust.

While the total surveyed area of about 70 km<sup>2</sup> may seem limited, the dataset spans a broad geographic range and includes diverse ice conditions. The 5467 analyzed images are non-overlapping and spaced at mostly regular intervals along the flight tracks, thereby keeping the observations statistically independent. Over 1.2 million individual ponds were identified in total, including 268,511 ponds larger than 1 m<sup>2</sup>. These provide a strong statistical foundation, particularly for small- to medium-sized ponds, which are the focus of this study. This extensive, varied dataset allows for a robust evaluation of the relationship between melt pond geometry and the presence of pressure ridges, and, to our knowledge, represents the first observational study capable of directly linking these two features at high resolution, while covering such a wide geographic range.

A more detailed history of the surveyed ice would further enhance our understanding of melt pond processes. While such data is often not available, targeted surveys focusing on specific ice regimes or revisits of the same ice area over a melt season would be favourable.

### 4 Conclusions





In this study, we analyzed coincident airborne image and laser altimeter data from three research flights north of Greenland to investigate the relationship between ridge fraction and melt pond properties. Specifically, we examined melt pond fraction, pond size distribution, and pond geometry and assessed how they are influenced by the presence of ridges. To ensure accurate pond identification, we applied a newly adapted brightness correction method to the aerial images, which automatically accounts for brightness gradients introduced by varying light conditions. To combine ridge and image data, we assigned a ridge fraction value to each image based on the corresponding laser altimeter data. This per-image classification allowed us to analyze how ridge fraction influences pond characteristics across the dataset.

While previous studies often associate smooth ice surfaces with high melt pond fractions, our results show that the connection between surface roughness and melt pond fraction is more complex. In particular, timing of the melt stages, such as melt onset and drainage events, play an important role. The temporal and spatial separation of the three flights presents challenges in

drawing general conclusions about the evolution of pond coverage under different ice conditions. Notably, we observe that high melt pond fractions are not exclusive to smooth or level ice (often associated with first-year ice) but can also develop on deformed multi-year ice with high ridge fractions. Recognizing this variability is essential for improving melt pond parameterizations in sea ice models, ensuring they accurately capture the diverse surface conditions observed in the field.





A central novel finding of this study is the dependence of pond size distribution and pond geometry on ridge fraction. Our analysis reveals that the presence of ridges affects the exponent of the power law function used to describe the pond size distribution. On relatively smooth ice, with a ridge fraction below 3.9%, this exponent is 1.4, while for rougher ice with a ridge fraction above 17.5%, the exponent takes on the value 1.51. This dependence of the power law exponent on ridge fraction demonstrates the influence of surface roughness on pond size distribution. Incorporating this relationship could be particularly useful in models that require a detailed representation of individual ponds or where an accurate pond size distribution is relevant for parameterization or as a reference score to test the validity of simulated pond geometry as demonstrated by Popović et al. (2018).

We find that the way ridges constrain the lateral expansion of ponds depends on the spatial scale: Small ponds exhibit more complex shapes in the presence of ridges, while large ponds are restricted in their complexity by the presence of ridges. This shift of behaviour behavior occurs around the same pond size scale as the shift in pond fractal dimension.

These results highlight the role of surface morphology in shaping melt pond characteristics, adding an important dimension to our understanding of pond evolution. As Arctic sea ice becomes younger and smoother, we expect shifts in the pond size distribution and the range of complexities reached by ponds. By incorporating the relationship between ridge fraction and pond properties, future modeling efforts could achieve a more accurate representation of melt pond characteristics, which is crucial for simulating sea ice albedo and meltwater storage. Further observational efforts, particularly those tracking the same ice over time and using high-resolution co-registrated images and digital elevation models, would help refine these findings and strengthen their applicability in models.

Code availability. For image classification, the PASTA-ice algorithm by Fuchs (2023b) was used. The Python code for the EM-Bird cable removal is available via https://gitlab.desy.de/helmholtz-imaging/awi\_cable. The Python code used for pressure ridge detection from laser altimeter data as described in Krumpen et al. (2025) is available via https://gitlab.awi.de/sitem/sbla\_processing.

Data availability. Brightness corrected images after cable removal, the corresponding cable mask, and the final georeferenced and classified images are available for download at https://doi.org/10.5281/zenodo.17222941. The laser altimeter data are available on PANGAEA (Krumpen et al., 2024a, b).

Author contributions. LB developed the concept and new methods used in this study, processed and analysed the image data, visualized and interpreted results and prepared the manuscript. TK, NN and MW supported conceptualization, methodology and interpretation. TK

processed the laser altimeter data and provided drift trajectories for the surveyed ice. GB and CH contributed to the concept of the study.

NF assisted with the methodology, most importantly the image classification. PH and OJ developed the cable removal algorithm. All authors

contributed by reviewing the manuscript.

Competing interests. CH is a co-editor-in-chief of The Cryosphere. The authors have no other competing interests to declare.

Acknowledgements. The authors thank Richard Gerum for his support with direct georeferencing using an adapted version of his Python package cameratransform. We also extend our gratitude to the Polar 6 crew, Kenn Borek Air, Station Nord and Alert, as well as the AWI technicians and logistics teams for their assistance during the airborne campaigns. The airborne campaigns were funded by the German Federal Ministry of Education and Research.

NF acknowledges funding from the Deutsche Forschungsgemeinschaft (DFG) under Germany's Excellence Strategy (EXC 2037; CLICCS – Climate, Climatic Change, and Society; project no. 390683824.

Paraphrasing tools were used in the preparation of this manuscript.

**Figure A1.** Flight tracks (circles) and full drift trajectories (small dots) of the surveyed ice, using IceTrack (see Section 2.8). The color represents the ice age.

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
