# Peer review of "Characterizing sea ice melt pond fraction and geometry in relation to surface morphology"

_EGUsphere, 2025_

## Referee Comment (RC1)

Review of: Buth et al., Characterizing sea ice melt pond fraction and geometry in relation to surface morphology

This study utilizes airborne imagery and altimetry captured on three flights over melting Arctic sea ice. The authors look to identify the link between melt pond fraction and the presence of sea ice ridges but find a complex relationship. The methodology is well described and the discussion topics were well chosen. It is a very interesting paper with a strong analysis. There are many minor comments and revisions for clarity and a few major points that would benefit from further analysis and/or longer discussion. Please find my general and specific comments below.

General:

The introduction reads like a list of references. It doesn't tell a cohesive story. Although the references included are good sources, I recommend a rewrite to make it flow better. Especially the paragraph starting at line 29- you flip back and forth between first year and multiyear ice and include landfast ice and it is all very confusing.

The discussion on pond geometry needs some clarification in the methodological description. How are ponds that intersect with image borders handled? What are the minimum and maximum pond sizes that can be observed in the flights at varying altitudes with the range of pixel sizes and images sizes. Perovich et al., 2002 has a good way to determine these values:

Perovich, D. K., Tucker III, W. B., & Ligett, K. A. (2002). Aerial observations of the evolution of ice surface conditions during summer. *Journal of Geophysical Research: Oceans*, *107*(C10), SHE-24.

Specific:

Line 6: what do you mean by high melt pond fraction (quantify).

Line 29: how do snow dunes affect pond formation- mentioned but not explained

Line 33: for the Eicken et al reference- how does the topography influence pond formation?

Line 34: Doesn't the presence of ridges on multiyear ice limit the spread of pond water as well? This is a confusing sentence

Line 40: which parameterizations in which models is this study referring to? This is a very broad statement.

Figure 1 caption: can you describe the source of the drift trajectories and indicate the way that they are marked (semi transparent dot marked every day?) And does the color correspond to the days since continuous melt onset at the date of the flight or does it change over the course of the drift trajectory? Potentially interesting information, but needs to be described better in the caption.

Table 1: does typical mean average or median or what?

Table 1: What does ground sample distance mean? It is not described in the text.

Figure 2: Can you use a different color for the cable lines in a/d/g and the altimeter measurements in b/e/h?

Line 88: what is a typical RGB sum that corresponds to the surface type? Can you give an average?

Line 100-102: I think this text is more valuable as a figure caption and does not need to be in the main text

Line 108: histogramm → histogram

Figure 3: this figure is a little blurry- I recommend increasing the dpi. The smoothed & filtered dots look more like a line, so I would just call them that in the figure caption

Line 146-150: Can you provide a few more details on how the time offset for the camera is refined based on the altimetry?

Line 152: I believe the references after e.g. should be listed in chronological order but check the style guide.

Line 160: can you remind the reader how many pixels $0.1 \text{ m}^2$ is?

Line 244: fewer → less

Line 246: are the dark/light ponds identified in the classification algorithm or is this a visual qualitative assessment? If included in the algorithm- can you provide numbers?

Figure 4: somewhere in the text can you describe why you think the observed SIC from images is so different from the OSISAF SIC?

Line 278: can you clarify? I think you mean that for the Western flight there is less time between the flight date and the melt onset date and that is why it is lower MPF?

Line 330: Snow depth and distribution is also a factor in melt pond distribution. It would be worth mentioning if not including a full discussion.

Line 370: for the circularity and general pond size distribution analysis, how do you handle the ponds that are on the edges of images? Are they eliminated from the analysis (via border intersection clearing)?

Figure 8. It would be useful to plot a common shape (circle) on the pond perimeter v pond size chart? Or note what it would look like (a straight line?)

Figure 8c. I don't understand what conceptualized and simplified visualization means? Is this just what you expected? Why? Explain in the text. The text currently does not clarify what this is.

Line 387: the description of the figure should be contained within the figure caption.

Line 405: this paragraph (point 3) is confusing. You say high perimeter to area ratio ponds form on more ridged ice across pond sizes. Then later say that this trend is reversed for large ponds and that when the perimeter to area ratio is high, it is on smooth ice. So the statement "broad range of pond sizes" needs to be clarified.

Line 408: Confusing comment about the fractal dimension- can you please clarify? Did you calculate the fractal dimension? With what metric? IF you are referring to another paper's finding please cite here.

---

## Author Comment (AC1)

This manuscript presents a study of the relationship between the surface morphology of sea ice (due to presence/absence of ridges) and two characteristics of melt ponding (areal pond fraction and individual melt pond circularity). The data, analysis, and results are described in a way that is comprehensive, clear, organized, and easy to read. I have only one minor suggestion (not a requirement) and a short list of very minor technical points. In my opinion, this manuscript can be published upon the authors' attention to these points.

> Thank you for your thorough review of our manuscript. Your comments have helped us improve the study. We are please to read that you support its publication. In the following, we address your comments one by one. Our replies are in blue. Line numbers are always referring to the first submitted version of the manuscript.

Suggestion (not required for publication!): In my opinion, a nice addition to this manuscript would be a figure containing a selection of images of ponds spanning a range of circularities. The description of pond circularity (how it is computed and what it represents) exists and is clear, so I don't think this information is missing. It simply occurred to me while reading the manuscript that a menu of examples of ponds with different circularities would be a really nice addition.

> We like this suggestion and we agree that visualizing the pond circularity will help to improve intuition on what circularity actually means. We have included some example pond shapes along with their circularity value as a new panel c in Figure 6. The ponds have been chosen such that they can also serve as examples for the pond size bins shown in Figure 8d.

[Figure]

Minor technical points:

4: coincident (also line 445)

> Changed to "coincident"

107: histogram (only one m)

    Changed to "histogram"

222: which on melt conditions (omit 'on')

    Removed "on"

244: fewer melt? How about 'less melt'?

    Changed to "less melt"

248: "...mix of ice types, including landfast first-year ice, secondyear ice, and multi-year ice." Would be helpful to know how it was discerned that all these types existed?

    We changed the respective sentences to the following to make it clearer that also the information on sea ice age used here is derived from the satellite backtracking in Section 2.8:

    "The Central flight (example shown in Fig. 4d) covered regions with a mix of ice types, including landfast first-year ice, second-year ice, and multi-year ice, as inferred from satellite backtracking of the ice drift (see Section 2.8 and Fig. A1). Much of this ice originated from the East Siberian Sea."

250: 'submerged ice'? Not sure exactly what this refers to? Rafted ice?

    Indeed, we had not defined "submerged ice" in this context yet. We added:

    "... submerged ice, here referring to parts of the ice floe that lie below the water surface, e.g. at floe edges, but are still visible in the nadir images"

259: "coastal areas" which coasts?

    We added: "..., in this case mostly the New Siberian Islands (see Fig. A1)"

261: "...slush-covered areas (example shown in Fig. 2g)." Am I supposed to be able to tell there is slush in the photo shown in Fig. 2g? If so, please help me know what to look for.

    The main purpose of including the figure reference here was to refer to an example image of the Eastern flight, as already done for the Western and Central flights. However, in this example there is actually some of the slush visible in the lower part of the image, see the zoomed in version below. We now guide the reader better to these areas by writing "example shown in Fig. 2g, e.g. roughly one-third up from the bottom of the image".

    Please note that we rephrased all instances of "slush" to the more descriptive and better defined wording "water-saturated snow", or in the case of "less slush" to "dryer snow".

[Figure]

275: "was varied", should just be "varied" (otherwise it sounds like it was manipulated)

Done

283: orange? More like gold on my screen

Changed to "gold" both in the text and in figure caption.

292: "very variable"? how about "highly variable"?

Changed to "highly variable"

391: "perimeter-area ratio" or perimeter:area ratio?

Changed to "perimeter-to-area ratio" to make it unambiguous and stay consistent throughout the text.

399: "previous subsection" how about "Section 3.3.1"?

Changed to "in Section 3.3.1"

423: "melt pond properties" how about stating it more specifically, as "melt pond geometry"? There are a lot of pond properties that are out of scope in this study. Here, "pressure ridges" are referred to, whereas the title says "surface morphology". I think maybe the title would be more fitting if it referred to "sea ice ridging"? Or even "ice surface morphology"?

You are absolutely right, there are so many other melt pond properties that we do not cover here! Here we are referring to both melt pond geometry and melt pond fraction, so in order to include both, we now changed the wording to "spatial melt pond properties".

We considered changing the title as you suggested, but we are hesitant because we would like to avoid double occurrence of the word "ice" in the title. Upon your and the editor's approval, we would like to keep the title as it is.

---

## Author Comment (AC2)

**General comments**

This manuscript describes the use of observations of sea ice and melt features and examines the relationships between the presence of melt ponds, ice type, ridging and age.

This manuscript addresses a critical and persistently under-quantified component of Arctic sea ice evolution: the relationship between surface morphology, e.g. ridge fraction, ice age, and the geometric characteristics of melt ponds. The authors use a combination of high-resolution optical imagery and coincident airborne altimetry to assess melt pond distribution and shape across a 70 km² section of ice north of Greenland. The paper's structure is clear and the results are well exposed and supported by the data analyzed.

I recommend the publication of this paper provided that the authors address a few points.

> Thank you for your thorough review of our manuscript. Your comments have helped us improve the study. We are please to read that you support its publication. In the following, we address your comments one by one. Our replies are in blue. Line numbers are always referring to the first submitted version of the manuscript.

**Specific comments**

The introductions contains all the right concepts, but it is often presented in a very fragmented way with very short phrases and paragraphs not properly linked to each other. The authors could improve it by grouping some of the very short phrases (e.g. lines 20-25), and making the links between the concept presented more explicit.

> Thank you for this feedback. We rewrote parts of the introduction and added some links between topics, while taking into account your and the other reviewers' comments. We have added the revised version of the introduction at the end of this document.

The methodology (between paragraphs 2.3-2.6), could be moved to an appendix and more details could be added. A much shortened version describing it could be added in the main manuscript.

> You bring up a valid point! We recognize that the Methods section takes up a lot of space in the manuscript. Before submitting the initial manuscript, we thoroughly thought about moving (parts of) the Methods to the Appendix, but we decided to include them in the main part of the manuscript because almost all of the subsections include new methodology and, in our opinion, are thus worth being incorporated into the main part. Upon your and the editor's approval, we would prefer to keep the structure as it is.

In particular, it would be nice to read about the choice of algorithm: here you use Fuchs (2023a). I am not questioning your choice, but considering that several other algorithm used for sea ice, I believe that the reasoning behind this choice should be more explicit.

> We completely agree that our reasoning for the choice of algorithm was missing. We now start Section 2.5 as follows:
>
> "For the semantic classification of the images into sea-ice surface types, we use the pixel-based classification algorithm by Fuchs (2023a), which uses a random forest classifier. It was particularly developed for the applied camera systems and allows for classification of aerial images collected under clear- and overcast skies. To ensure compatibility with our dataset after the modified brightness correction (see Section \ref{sec:brightness}), we expanded the original training dataset with pixels sampled from our images and retrained the model. While a variety of sea ice surface classification for airborne images is available (e.g. Buckley et al., 2020; Wright and Polashenski, 2018), we selected this algorithm due to its tailored design for the imaging system used in our campaigns. It provides ..."

Another study (Buckley et al., 2020), uses pixel classification and seems to have less limitations when sea ice concentration is low.

> Please note that low sea ice concentration in individual images is no problem at all. It can become a problem for the semi-automatic image brightness correction (Section 2.2), but only if the low sea ice concentration persists throughout large parts of the flight. The classification algorithm itself is not affected by the sea ice concentration. See also the next comment for more details effect of sea ice concentration on the brightness correction.
>
> In case you a referring to problems related to submerged ice: As mentioned in Section 3.2.3, a low sea ice concentration often introduces more brash and submerged ice. By its nature, a pixel-based approach on its own can not distinguish between submerged ice and a melt pond. This can introduce misclassification, in our case this is only a small fraction of the total melt pond class. Both Fuchs (2023a) and Buckley et al. (2020) report on misclassification due to submerged ice.

My main concern is about the amount of data analysed: It would be nice to have a feel of the total number of images acquired and how many were used: how many images were discarded because of too low SIC for example? What is the threshold assumed to discard an image (it could be more explicit I line 92-93).

> Thank you for bringing this up. We realize that our wording is not precise enough, as it might sound like we can not use images with low SIC at all. As a matter of fact, all 5467 images mentioned in Table 1 are used for subsequent melt pond analysis, just not all of them contribute to the brightness assessment (see Figure 3). We changed the text to the following:
>
> "Images dominated by open water or melt ponds are filtered out from the brightness assessment step, as explained below. These images are not discarded but are assigned a

brightness value interpolated from neighboring images and are fully retained for all
subsequent processing steps. It should be noted that this brightness correction method is not
well suited for regions with many such images, such as the marginal ice zone (MIZ) with low
sea ice concentration. "

A more thorough description of the whole campaign would be useful (there is a map in figure A1, but
there is no time described).

Thank you for raising this point, as indeed the timing of the survey flights is only mentioned in
Table 1. For a better overview, we included the dates (along with the names we gave to the
flights) in the campaign description in Section 2.1. Figure A1 shows the same flight tracks as
Figure 1 but with the entire drift trajectories to show the source areas of the investigated sea
ice including its age. This is mentioned in the first sentence of Section 3.1. We now added the
information that it also shows sea ice age of the surveyed ice, to avoid confusion with Figure 1.

In line 113 you mention that there is no need to find references for the brightness temperature more
than once per flight. Is this due to the length of the flight, to the weather (see also line 79).

Our wording was not ideal here. We are referring to the method itself, which makes sure that
reference areas from images across the whole brightness spectrum are being used. We write
this more explicitly now. We also changed the phrasing to make more clear that this means
finding reference areas for a total of 10 images per flight. We now write:

"The described method ensures that the selected images used for calibration span the full
range of brightness conditions encountered during a flight. As each image is assigned to one of
10 brightness categories, we determined that it is sufficient to identify reference areas in just
one representative image per category, resulting in a total of 10 images assessed per flight.

This revised approach significantly reduced the manual labor involved in analyzing optical
imagery from long flights over Arctic sea ice, relying solely on the images themselves."

In paragraph 3.4, Could you please add a comment on the reliability of the statistics: the area covered
by the three flights (70km2), is not huge. Could you make a comment on how representative your
conclusions are? When devising a new melt pond parameterization this would be an important
information to consider.

We agree that we should include more information about this. We added the following
paragraph to Section 3.4, after line 440:

While the total surveyed area of ~70 km² may seem limited, the dataset spans a broad
geographic range and includes diverse ice conditions. The 5467 analyzed images are non-
overlapping and spaced at mostly regular intervals along the flight tracks, thereby keeping the
observations statistically independent. Over 1.2 million individual ponds were identified in
total, including 268,511 ponds larger than 1 m². These provide a strong statistical foundation,

particularly for small- to medium-sized ponds, which are the focus of this study. This extensive, varied dataset allows for a robust evaluation of the relationship between melt pond geometry and the presence of pressure ridges, and, to our knowledge, represents the first observational study capable of directly linking these two features at high resolution, while covering such a wide geographic range.

**Technical corrections**

Line 4: "coincided laser altimeter data": correct to "coincident laser altimeter data"

Changed to "coincident".

Line 44-46: too many interruptions for refs: please move them to the end of the phrase if the journal allows it.

Done.

Caption - figure 2: "examplary images": correct to "example of images". Also in line 480.

Done.

The pink circles in this figure are either too big, too close or too many: they look like a line.

The individual measurements are so close together that they can be difficult to distinguish. We now use short horizontal lines, a compromise between making sure the markers are big enough to be seen at all and thin enough to be seen individually.

Line 68 – 70: subsections 2.2 on brightness correction and 2.3 on the removal of the tow cable introduce novel approaches and are therefore described in detail. Pls review punctuation for clarity.

Done.

Line 84: " by determining the modal value of the sum of the R, G, and B channel rightness per image (modal RGB sum), as..': in the first part of the phrase pls write in full Red, Green and Blue.

Done.

Line 100: "...along each flight track" (if multiple are referred to) rather than "Along the flight track".

We agree that this wording might have caused confusion. As suggested by Reviewer #1, we omit the description of the figure at this location in the main text, as this kind of description belongs in the figure caption. The sentence you are referring to is thus not included in the manuscript any more. In the figure caption we explicitly state that these are the images along the Western flight track.

Line 102: what is an expected modal RGB sum value?

Thank you for raising this point as indeed this was a bit unclear in the previous version of the manuscript. As the modal RGB sum is varying along track, there is no typical or expected value for the snow/ice class. Reviewer #1 had a similar comment. To account for both we included the following text in the Section 2.2, in line 88 of the previous manuscript:

"However, the RGB sum of each surface type varies significantly along each flight, depending on the prevailing brightness conditions. For instance, values for snow and ice during the Western flight range from below 20,000 to over 50,000 (see Fig. 3). Given this variability, there is a need for a relative rather than a fixed reference. That is why the following approach does not rely on a single expected brightness value for snow or ice, but instead dynamically adapts to observed conditions and thereby ensures consistent correction across a broad spectrum of light conditions."

Line 107: "histogramm stretch approach": correct to "histogram stretch approach"

Changed to "histogram".

Line 126: First rather than firstly

Changed to "First"

Line 146: the camera timestamps are precise only to 1 second": pls consider rephrasing to "...are only precise to one second".

Changed to "... are only precise to one second"

Line 188 and following: Could you please explain the 3.9% and 17.5% more clearly? Why these numbers?

Thank you for pointing this out. These values correspond to the quantiles mentioned in the sentence before. We added a sentence after the one mentioning the threshold values to make this more clear. The paragraph now reads:

"In other cases, the classified images were grouped into three ridge fraction categories: low, medium, and high, based on quantiles of the observed ridge fraction distribution across all images. This means the lowest third of the distribution, comprising ridge fractions below 3.9%, is categorized as "smooth ice", while the highest third, comprising ridge fractions above 17.5%, is categorized as "rough ice". Specifically, these thresholds represent the 33rd and 67th percentiles of the observed ridge fraction range, respectively."

Line 235: the colon is probably not appropriate. Pls also consider to have three subsections for the three chosen flights.

We replaced the colon by a period, but decided to keep the three flights in one subsection, in order to keep the flow of the text.

Line 244: fewer?

Changed to "less melt"

Line 338: Perhaps the ref to Horvat 2020 should come to the end of this phrase, since their study is based on the hypothesis that the distribution of pond is of fundamental importance for the distribution of light and energy under the ice.

Thanks for spotting this. We moved the reference to the end of the next sentence, as suggested.

Line 123, 374 and 419: Capitals after colons in a few instances. Please correct throughout

We agree that it is common to start with a lowercase letter after a colon in British English. However, for other ambiguous spellings, we opted for American English, where, in our understanding, it seems to be common to start with an uppercase letter if the colon is followed by a complete sentence. To stay consistent throughout the manuscript, we use an uppercase letter here. We are happy to change this upon request according to editorial guidelines.

Others:

"preflight" and "postflight" not consistently hyphenated: pls correct in a uniform way of choice.

We assume you are referring to "pre-melt". We have made sure that it is spelled consistently throughout the manuscript.

"behaviour" vs. "behavior": please ensure consistency throughout.

Changed to "behavior" in all instances throughout the manuscript

**Revised version of the introduction:**

[revised manuscript text omitted]

---

## Author Comment (AC3)

Review of: Buth et al., Characterizing sea ice melt pond fraction and geometry in relation to surface morphology

This study utilizes airborne imagery and altimetry captured on three flights over melting Arctic sea ice. The authors look to identify the link between melt pond fraction and the presence of sea ice ridges but find a complex relationship. The methodology is well described and the discussion topics were well chosen. It is a very interesting paper with a strong analysis. There are many minor comments and revisions for clarity and a few major points that would benefit from further analysis and/or longer discussion. Please find my general and specific comments below.

> Thank you for your thorough review of our manuscript. Your comments have helped us improve the study. We are please to read that find it interesting. In the following, we address your comments one by one. Our replies are in blue. Line numbers are always referring to the first submitted version of the manuscript.

General:
The introduction reads like a list of references. It doesn't tell a cohesive story. Although the references included are good sources, I recommend a rewrite to make it flow better. Especially the paragraph starting at line 29- you flip back and forth between first year and multiyear ice and include landfast ice and it is all very confusing.
> Thank you for this feedback. We rewrote parts of the introduction and added some links between topics, while taking into account your and the other reviewers' comments.
> In particular, we have changed the structure of the paragraph formerly starting in line 29, as suggested. We believe that with the new order, the paragraph is more intuitive and easier to understand. We have added the revised version of the introduction at the end of this document.

The discussion on pond geometry needs some clarification in the methodological description. How are ponds that intersect with image borders handled? What are the minimum and maximum pond sizes that can be observed in the flights at varying altitudes with the range of pixel sizes and images sizes. Perovich et al., 2002 has a good way to determine these values:
Perovich, D. K., Tucker III, W. B., & Ligett, K. A. (2002). Aerial observations of the evolution of ice surface conditions during summer. Journal of Geophysical Research: Oceans, 107(C10), SHE-24.
> We agree that this is an important topic to include.
> We added the following text to the methods, in section 2.5, now named "Surface classification and geometric considerations", after line 160:
> "While the minimum observable pond size is defined by this threshold, the maximum size is limited by the image dimensions. Due to the relatively low flight altitude required for sea ice thickness retrieval, the classified images are small, and larger ponds are increasingly likely to extend beyond image boundaries. As stated by Perovich et al. (2002), the probability that a circular pond with radius R is fully contained within an image of width W and length L is given by:

$$p(R) = \frac{(L - 2R)(W - 2R)}{LW + 2LR + 2WR + \pi R^2}$$

Using this equation, we estimate that the probability of fully capturing a circular pond with an area of 50 m² —a threshold that will be relevant relevant in our analysis— is about 76% for the Western flight and 74% for the Central and Eastern flights. As pond size increases, this probability decreases, and larger ponds are more likely to be cut off at image edges. As a result, the dataset predominantly captures smaller ponds in full, while larger ones may only appear partially.  In such cases, we treat ponds intersecting image borders as if they were whole. Consequently, our observations are inherently biased toward smaller ponds, and this constraint shapes the scope of our analysis, focusing on small to medium-sized melt ponds."

In Section 3.3.1 about the ponds size distribution, we already discuss the introduced bias towards smaller ponds.

We account for the effect on pond circularity in Section 3.3.2, by adding the following text in line 377:
"Additionally, ponds that are only partially captured within an image may appear less geometrically complex than they actually are, as portions of their shape are not visible. This phenomenon is most evident in larger, more irregular ponds, where the removal of branches or extensions can lead to a reduction in the perceived complexity and in particular on the calculated circularity value. For ponds of smaller, rounded proportions, the impact is negligible. The ponds are already not perfectly circular due to the pixel-based classification process; therefore, an image edge does not significantly alter their shape. The intersection of ponds with image edges thus introduces a bias towards rounder ponds, corresponding to lower circularity values.
Nevertheless, within the context of this study, the same methodology has been consistently employed, and the same limitations apply, allowing for valid comparisons of circularity among ponds within our dataset."

Specific:
Line 6: what do you mean by high melt pond fraction (quantify).
Indeed, it was not clear what we mean by "high" melt pond fraction. What is considered high depends on many aspects, including in our case which flight we are looking at (see Fig. 6a). With the sentence we wanted to express that pond fractions can be as high on ridged ice as they are on smooth ice, we therefore changed the respective sentence to:
"Our results reveal that melt pond fractions on heavily deformed multi-year ice can reach values comparable to those on smooth ice, with similar distributions observed for both."
This should make more clear what we want to emphasize.
We also considered writing an absolute number e.g. "higher than 30%" (see histogram in Fig. R1.1 below), but we think this might be more confusing to the reader, as they might compare it to Figure 6a, where such high numbers are rare due to the used resampling method.

[Figure]

*Figure R1.1: Histogram of melt pond fraction per image for smooth and rough ice, as defined in Section 2.7 of the manuscript, before resampling and including data from all three flights. (Figure not included in manuscript)*

Line 29: how do snow dunes affect pond formation- mentioned but not explained

We rewrote this part to better explain how meltwater collects in low-lying areas between the elevated features such as snow dunes. See attached revised introduction for details.

Line 33: for the Eicken et al reference- how does the topography influence pond formation?

Thank you for the question. We wanted to mention the influence of previous melt seasons here, with some melt ponds reappearing at the same location as in the previous summer. Your comment made us realize that we accidentally referenced a different Eicken paper, it should be Eicken et al. (2001). We corrected this and added two more references that fit the topic. For details see revised introduction.

Line 34: Doesn't the presence of ridges on multiyear ice limit the spread of pond water as well? This is a confusing sentence

Yes, it limits the spread, but this does not necessarily mean that it limits pond fraction as well. In the cited study, this difference between FYI and MYI is mentioned. We believe that this aspect is now better presented in the revised introduction.

Line 40: which parameterizations in which models is this study referring to? This is a very broad statement.

We now explicitly mention these specifications when citing the study. See attached introduction for details.

Figure 1 caption: can you describe the source of the drift trajectories and indicate the way that they are marked (semi transparent dot marked every day?) And does the color correspond to the days since continuous melt onset at the date of the flight or does it change over the course of the drift trajectory? Potentially interesting information, but needs to be described better in the caption.

Thank you for pointing this out. The NASA Arctic Sea Ice Melt product provides the first day of continuous melt as a "day of year" value on a grid. Thus we cannot use it to determine the exact number of continuous melt days that a drifting region has experienced along its path. For this reason, we calculated the average number of days along the shown summer drift path. The

color represents this average value with respect to the day of the survey; therefore, it is constant within each trajectory. The approach to derive this value is also described in Section 2.9. We changed the figure caption to the following:

"Map showing the flight tracks (circles) of the Western (W), Central (C), and Eastern (E) flights, along with the satellite-derived ice drift trajectories for the 70 days prior to each survey (Krumpen et al., 2018). Trajectories are shown as semi-transparent dots, with one dot per day. Flight tracks and the corresponding drift paths are color-coded by the average number of continuous melt days along each trajectory until the survey time, based on the NASA Arctic Sea Ice Melt product of the respective year (2018 for W, 2016 for C and E)."

Table 1: does typical mean average or median or what?

Thanks for catching this! We indeed mean "mean image size" and "mean ground sample distance". We corrected it in both instances.

Table 1: What does ground sample distance mean? It is not described in the text.

We added a short explanation to the paragraph starting in line 64 of the original manuscript. The paragraph now reads:

"Altitude variations during the measurements were minimal, mostly within 2.5\,m, with maximum variations of about 5\,m. These differences in altitude, together with the fixed camera specifications, affect the ground sample distance (GSD), defined as the distance between adjacent pixel centers on the ground, which in turn determines the surface area covered by each image. As a result, the surface area per image varies slightly throughout and, more noticeably, between the flights."

Figure 2: Can you use a different color for the cable lines in a/d/g and the altimeter measurements in b/e/h?

Done.

Line 88: what is a typical RGB sum that corresponds to the surface type? Can you give an average?

Thank you for raising this point as indeed this was a bit unclear in the previous version of the manuscript. As the modal RGB sum is varying along track, there is no typical or expected value for the snow/ice class. Reviewer #2 had a similar comment. To account for both we included the following text in the Section 2.2, in line 88 of the previous manuscript:

"However, the RGB sum of each surface type varies significantly along each flight, depending on the prevailing brightness conditions. For instance, values for snow and ice during the Western flight range from below 20000 to over 50000 (see Fig. 3). Given this variability, there is a need for a relative rather than a fixed reference. That is why the following approach does not rely on a single expected brightness value for snow or ice, but instead dynamically adapts to observed conditions and thereby ensures consistent correction across a broad spectrum of light conditions."

Line 100-102: I think this text is more valuable as a figure caption and does not need to be in the main text

> Indeed, we already describe this in the figure caption and don't need to do so again in the main text. As suggested, we now omit the figure description in (formerly) line 100 and only leave the sentence "Figure 3 illustrates the process of assigning brightness values and categories to each image."

Line 108: histogramm → histogram

> Changed to "histogram"

Figure 3: this figure is a little blurry- I recommend increasing the dpi. The smoothed & filtered dots look more like a line, so I would just call them that in the figure caption

> We increased the dpi to 600 and changed "circles" to "line".

Line 146-150: Can you provide a few more details on how the time offset for the camera is refined based on the altimetry?

> Sure! The paragraph now reads:
> "To address this issue, a two-step approach is employed. Initially, the data are matched with 1-second precision. Then, using surface profiles from the laser altimeter, the time offset for the camera is refined manually for each flight. This is done by visually identifying and matching distinctive features, such as pressure ridges and smooth ice areas, that appear in both the altimeter data and the images. By comparing the spatial alignment of multiple such features, a spatial offset is determined, which can be converted to a temporal offset. The georeferencing algorithm is then rerun using this refined time offset to improve the spatial accuracy. The results are verified by re-examining multiple locations along each flight track."

Line 152: I believe the references after e.g. should be listed in chronological order but check the style guide.

> Done.

Line 160: can you remind the reader how many pixels 0.1 m^2 is?

> We rephrased the sentences to the following:
> "Following the classification process, neighboring pixels of the same class are combined into objects. To reduce noise, objects smaller than 100 pixels, here corresponding to an area of approximately 0.1m², are filtered out, as applied by Huang et al. (2016) and Fuchs et al. (2024). This process ensures that the classification results are robust and relevant to the scale of our analysis."

Line 244: fewer → less

> Changed to "less melt"

Line 246: are the dark/light ponds identified in the classification algorithm or is this a visual qualitative assessment? If included in the algorithm- can you provide numbers?

> Thank you for this question, we agree that we should have mentioned this in the text. We are here referring to a visual qualitative assessment. The subclasses "bright pond" and "dark pond"

exist in the classification algorithm (Fuchs 2023a), but we have not checked their performance in combination with the new adapted brightness correction and this dataset, nor do we use these subclasses in the later analysis. For other uses cases, it could certainly be insightful to make use of these and other available subclasses.

We added "Based on visual qualitative assessment of the images, …" to the sentence.

Figure 4: somewhere in the text can you describe why you think the observed SIC from images is so different from the OSISAF SIC?

Thank you very much for this question, it made us recalculate the SIC from the airborne imagery. Upon recalculation we realized that we had made a mistake: Instead of the sea ice concentration, we were showing the fraction of area classified as snow/ice over the total image area, not taking into account that the "pond" areas need to be included here as well. Naturally, this led to SIC values that are too low. The correct way to calculate SIC per image from the classification results is:

$SIC = (\ a_{ice} + a_{pond}\ ) /\ a_{total}$

or even $SIC = 1 - (a_{ow} / a_{total})$

, with $a_{total} = a_{ice} + a_{pond} + a_{ow}$

, where $a_{ice}$ , $a_{pond}$ and $a_{ow}$ denote the areas of the image covered by snow/ice, pond/submerged ice and open water, respectively.

With the correct approach, we obtain the following new mean SIC values per flight:

Western flight: 0.93

Central flight: 0.83

Eastern flight: 0.91

We have updated all instances of the SIC values in the table, the main text and Figure 4. The corrected SIC values are much closer to the mean OSI SAF values now, and all fall within their standard deviation.

[Figure]

Please note that we have also changed the order in which the flights appear in the legend to make it consistent with the order typically used throughout the manuscript.

Line 278: can you clarify? I think you mean that for the Western flight there is less time between the flight date and the melt onset date and that is why it is lower MPF?

Indeed, this was not phrased ideally. We intended to say that this "less time" certainly plays a role, but in particular we wanted to emphasize that it is not the only factor determining melt pond fraction. To clarify, we rephrased it as follows:

"In comparison, the melt pond fraction observed during the Western flight is lower, with an average of 0.13. This flight was conducted in a region where the melt onset occurred later and closer in time to the survey date, reducing the time available for pond formation and evolution before the overflight. However, the melt onset date alone does not determine the pond fraction at any given time. While it provides context for the visual differences in ponding during the three flights, additional factors influence pond development. Throughout the four stages of melt defined by Eicken et al. (2002), melt pond coverage evolves continuously with periods of expansion, drainage, and potential refreezing. The observed melt pond fractions in our data reflect this complex temporal variability of ponds."

Line 330: Snow depth and distribution is also a factor in melt pond distribution. It would be worth mentioning if not including a full discussion.

We agree to this point, but can not distinguish between different surface types from laser altimetry. To make this clear, we chose to add some sentences to the methods and discussion sections.

In the Methods, Section 2.6, after line 173, we include the following:
"Such features may be composed of ice and/or accumulated snow; however, we refer to them as ridges throughout this study, as pressure ridges are the dominant elevated features on the sea ice during these summer surveys."

We now also mention this again in the Discussion, Section 3.4, after line 432:
"Additionally, our ridge detection method does not allow us to distinguish between snow and ice in the detected surface features. As a result, elevated structures identified from the laser altimeter are referred to as ridges throughout this study, although they may in some cases include contributions from snow accumulation. This limitation should be kept in mind, as snow depth and distribution can influence early meltwater pooling and thereby the location and development of melt ponds."

Line 370: for the circularity and general pond size distribution analysis, how do you handle the ponds that are on the edges of images? Are they eliminated from the analysis (via border intersection clearing)?

See above.

Figure 8. It would be useful to plot a common shape (circle) on the pond perimeter v pond size chart? Or note what it would look like (a straight line?)

Thank you for the suggestion. We have added the line showing the perimeter-to-area of a circle to Figure 8b.

Figure 8c. I don't understand what conceptualized and simplified visualization means? Is this just what you expected? Why? Explain in the text. The text currently does not clarify what this is.

We agree that this was not sufficiently explained in the previous version of the manuscript, and that the word "simplified" and the arrowhead in the figure were misleading in this context. Our approach was to split up the rather complex Figure 8a into different conceptual figures (panels b, c and d) that each show one of the aspects we explain in the points 1 to 3 in the text. We now write this explicitly when introducing Figure 8:

"To better understand the influence of ridges on pond shapes and to clarify the observed differences among the various flights, we present Figure 8. Panel a shows the distribution of melt pond areas and perimeters for every individual pond on a binned grid. Several key trends emerge from this figure, each represented by one subsequent panel of this figure: …"

Out of these panels, panel c is meant to visualize the background trend, or in a way the "lower frequency mode" of Figure 8a, so the relationship of ridge fraction observed for ponds with both increasing size and perimeter. (Panel d could be interpreted as the "higher frequency mode".) We changed the caption of panel c to:

"Conceptualized illustration of the general trend of decreasing ridge fraction with increasing pond size and perimeter, as seen in panel a."

In the main text, we changed it to:

"2. A clear second trend is evident in Figure 8a and conceptually visualized in Figure 8c, which illustrates the background mode of the dependency between pond geometry and surface roughness: As both pond area and perimeter increase, the ponds tend to appear on ice with, on average, lower ridge fractions. "

Line 387: the description of the figure should be contained within the figure caption.

Agreed. As this information is already included in the figure caption, we simply removed these sentences (lines 387-389) from the main text.

Line 405: this paragraph (point 3) is confusing. You say high perimeter to area ratio ponds form on more ridged ice across pond sizes. Then later say that this trend is reversed for large ponds and that when the perimeter to area ratio is high, it is on smooth ice. So the statement "broad range of pond sizes" needs to be clarified.

We agree that this wording was not precise enough. We now add a specification:

"… across a broad range of pond sizes, from the smallest detected ponds up to about $10^2 \ m^2.$"

Line 408: Confusing comment about the fractal dimension- can you please clarify? Did you calculate the fractal dimension? With what metric? IF you are referring to another paper's finding please cite here.

Thanks for pointing out this missing information. We did not calculate the fractal dimension, instead we are referring to other studies and the shift in trend in Fig. 8b. To make this clear, we now write:

[revised manuscript text omitted]